# Transition-metal-free silylboronate-mediated cross-couplings of organic fluorides with amines

Jun Zhou [1], Zhengyu Zhao[1] & Norio Shibata [1,2] ✉

C–N bond cross-couplings are fundamental in the field of organic chemistry. Herein, silylboronate-mediated selective defluorinative cross-coupling of organic fluorides with secondary amines via a transition-metal-free strategy is disclosed. The cooperation of silylboronate and potassium *tert*-butoxide enables the room-temperature cross-coupling of C–F and N–H bonds, effectively avoiding the high barriers associated with thermally induced $S_N2$ or $S_N1$ amination. The significant advantage of this transformation is the selective activation of the C–F bond of the organic fluoride by silylboronate without affecting potentially cleavable C–O, C–Cl, heteroaryl C–H, or C–N bonds and $CF_3$ groups. Tertiary amines with aromatic, heteroaromatic, and/or aliphatic groups were efficiently synthesized in a single step using electronically and sterically varying organic fluorides and *N*-alkylanilines or secondary amines. The protocol is extended to the late-stage syntheses of drug candidates, including their deuterium-labeled analogs.

F-containing organic compounds are frequently prepared by pharmaceutical and agrochemical researchers because incorporating F atoms can finely adjust the chemical and metabolic stabilities, lipophilicities, and acidities/basicities of these species[1–3]. In medicinal aspects, methodologies and reagents used to synthesize organofluorine compounds efficiently have been intensely developed over recent decades[4–7], and more than 350 fluoro-pharmaceuticals[8] and 430 fluoro-agrochemicals[9] are registered. Organofluorine compounds are abundant and readily accessible, rendering them attractive as functional moieties and as building blocks for use in further organic synthesis. However, the chemical conversions of fluorinated moieties to other functional groups are very challenging[10–15]. Compared with C–I/Br/Cl bonds, the C–F bond is relatively inert and has the highest bond dissociation energy. In addition to the high reaction temperatures required in C–F bond cleavage, a strong base is necessary to facilitate the transformation.

Aromatic tertiary amine moieties are critical structural features of molecules used in pharmaceuticals, agroscience, bioactive natural products, and materials science (Fig. 1a)[16–22]. To date, the most reliable preparation methods of aromatic tertiary amines are the transition-metal-catalyzed C(sp²)–N couplings of aryl (pseudo)halides with amine nucleophiles, such as the Ullmann coupling[23,24], the Buchwald–Hartwig reaction[25–27], and metallaphotoredox amination[28–30] (Fig. 1b). Although efficient syntheses of aromatic tertiary amines are among the top five reactions performed globally to synthesize high-value products[31], green chemistry requires the development of transition-metal-free systems. Generally, transition-metal-free aminations of aryl (pseudo) halides have limitations such as low regioselectivities and the requirement of a strong base and high temperature[32–39]. Moreover, inert C–F-containing organic fluorides are seldom used in such C–N couplings, particularly under mild conditions, owing to the high bond dissociation energies of the C–F bonds. Although the defluorinative aminations of aryl fluorides have gained attention, the reported methods require strong bases, transition metal catalysis, harsh conditions, or activation of aromatic rings with strong electron-withdrawing groups[10–15]. Besides, strong base-mediated amination reactions of aromatic C–F bonds provide the amination products as mixtures of regio-isomers, as these reactions proceed via benzyne intermediates[32–34]. In 2015, Cao, Shi et al. reported a direct $S_NAr$ reaction of aromatic fluorides by aromatic amines using ${}^tBuOK$ in DMSO[35].

[1]Department of Nanopharmaceutical Sciences, Nagoya Institute of Technology, Gokiso, Showa-ku, Nagoya 466-8555, Japan. [2]Department of Life Science and Applied Chemistry, Nagoya Institute of Technology, Gokiso, Showa-ku, Nagoya 466-8555, Japan. ✉e-mail: nozshiba@nitech.ac.jp

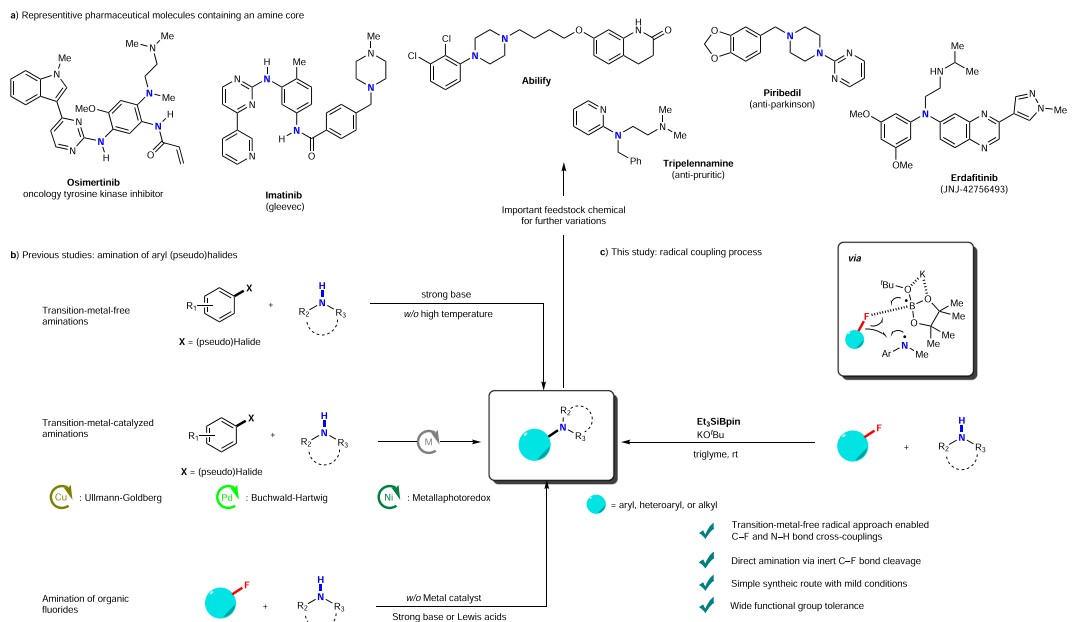

**Fig. 1 | Cross-couplings and related reactions of aryl or alkyl halides with secondary amines. a** Representative pharmaceutical molecules containing aryl and alkyl amines. **b** Previous studies. **c** This study.

Diness et al. disclosed the $S_NAr$ reaction of aromatic fluorides with alkyl amines using LiHMDS[36]. However, both methods require a high reaction temperature of 90–100 °C, except for the selected perfluorinated benzenes[35,36]. Furthermore, the $S_NAr$ reaction of perfluorinated benzenes with pyrroles by NaH[37] and N-heterocycle-assisted $S_NAr$ reaction of *ortho*-heterocyclic aryl fluorides with aryl amines using LiH were reported[38], 110–153 °C temperature conditions are required. Recently, the amination of electron-rich aryl fluorides was achieved under the 20–50 °C using photocatalysts and blue LEDs; however, such methods require electron-rich moieties, such as the OMe group, on the aromatic moieties, to activation of the substrates[40,41] Thus, these methods are not useful for electron-poor aryl fluorides, in contrast to the traditional $S_NAr$ reaction of aryl fluorides, which favors electron-poor substrates. Aminations involving the cleavage of C(sp³)–F bonds of alkyl fluorides are also a challenging issue, and the protocols are hitherto mostly limited to active benzyl or allylic fluorides, with strong Lewis acids, such as La[N(SiMe₃)₂]₃ and YbI₃, generally required[42–45].

As a continuation of our studies regarding C–F bond functionalization[46–48], we herein report the transition-metal-free defluorinative aminations of aryl fluorides using secondary amines, which proceed in the presence of triethylsilylboronate (Et₃SiBpin) and potassium *tert*-butoxide (KOᵗBu, Fig. 1c). A wide variety of aryl fluorides react with secondary acyclic and/or cyclic N-alkylanilines and/or dialkylamines at room temperature. This reaction affords aromatic tertiary amines in good-to-excellent yields via the cleavage of a C(sp²)–F bond in the presence of Et₃SiBpin, without employing transition-metal catalysis or a photoredox system. The (hetero)aryl fluorides easily react with unsubstituted N-alkylanilines and substituted N-methylanilines, enabling the syntheses of numerous structurally varied aromatic tertiary amines that regioselectively incorporate aryl, heteroaryl, and/ or alkyl groups at the N centers. Defluoroamination progresses with high regio- and chemoselectivity. The potentially cleavable C–O bonds of ethers, C–Cl and C–CN bonds, and C(sp²)–H bonds of heteroaromatic compounds are unaffected. Additionally, the C(sp³)–F bonds of the CF₃ and OCF₃ groups are unaffected. Notably, despite the unaffected CF₃ groups, this cross-coupling protocol is extended to the reactions of alkyl fluorides containing C(sp³)–F bonds with secondary amines to yield defluoroamination products with newly formed C(sp³)–N bonds.

## Results and discussion
### Silylboronate-mediated cross-coupling reactions of organic fluorides and N-alkylanilines
Initially, 4-fluorobiphenyl (**1a**) and N-methylaniline (**3a**) were used as model substrates to investigate defluoroamination. The desired product, N-methyl-N-phenyl-4-biphenylamine (**4aa**), was obtained in a yield of 41% under specific conditions [Et₃SiBpin (1.5 equiv.) and KOᵗBu (2.5 equiv.) in diglyme at room temperature, entry 1, Table 1]. Although higher amounts of Et₃SiBpin (2.0 equiv.) and KOᵗBu (4.0 equiv.) insignificantly improved the yield (44%, entry 2), the use of 3.0 equiv. of **3a** increased the yield to 58% (entry 3). Remarkably, the solvent affected the outcome of the defluoroamination. A yield of 81% of **4aa** was obtained under the same conditions as that of entry 3 but in triglyme (entry 4). When the reaction was carried out in THF, the yield significantly decreased to 5% (entry 5); however, it increased to 75% in the presence of 18-crown-6 (entry 6). The defluoroamination proceeded effortlessly when the reaction time was extended to 24 h in triglyme, generating a yield of 91% (88% isolated, entry 7). Control experiments revealed the necessity of using Et₃SiBpin and KOᵗBu (entries 8 and 9). Slight modifications to the reactant ratio and those of KOᵗBu and Et₃SiBpin did not improve the yields (entries 10–14). We confirmed the reproducibility of the optimal result (entry 7) by completing the reaction at double the scale to produce **4aa** in a 93% yield (89% isolated, entry 15). A gram-scale reaction was also performed using 5.0 mmol of **1a** to demonstrate the reproducibility of the reaction (1.13 g, 87%). We further examined the reaction using different silylboronates (ᵗBuMe₂SiBpin, PhMe₂SiBpin, (Me₃Si)₃SiBpin). Interestingly, the yield decreased as the steric hindrance of the silylboronates increased (entries 16–18), and some amount of the starting material (**1a**) remained unreacted. Further details on the optimization of the reaction conditions are shown in the Supplementary Information (Supplementary Tables 1–5).

### Scope and limitations
The substrate scope of this silylboronate-mediated direct amination was further evaluated using the optimal reaction conditions (entry 7, Table 1). As shown in Fig. 2, aryl fluorides **1** with various electronic properties were reacted with **3a**. First, four types of π-extended aryl fluorides (**1a–1d**), including the sterically hindered *ortho*-substituted

**Table. 1 | Optimization of the defluoroamination conditions[a]**

| Entry | 3a | Si–B | KO$^t$Bu | Solvent | Time (h) | 4aa (%)[b] |
|---|---|---|---|---|---|---|
| 1 | 1.5 | 1.5 | 2.5 | diglyme | 8 | 41 |
| 2 | 1.5 | 2.0 | 4.0 | diglyme | 8 | 44 |
| 3 | 3.0 | 2.0 | 4.0 | diglyme | 8 | 58 |
| 4 | 3.0 | 2.0 | 4.0 | triglyme | 8 | 81 |
| 5 | 3.0 | 2.0 | 4.0 | THF | 8 | 5 |
| 6 | 3.0 | 2.0 | 4.0 | THF with 18-crown-6 (4.0 equiv.) | 8 | 75 |
| 7 | 3.0 | 2.0 | 4.0 | triglyme | 24 | 91 (88) |
| 8 | 3.0 | – | 4.0 | triglyme | 24 | 0 |
| 9 | 3.0 | 2.0 | – | triglyme | 24 | 0 |
| 10 | 3.0 | 2.0 | 2.0 | triglyme | 24 | 7 |
| 11 | 3.0 | 2.0 | 5.0 | triglyme | 24 | 88 |
| 12 | 3.0 | 1.5 | 4.0 | triglyme | 24 | 80 |
| 13 | 1.5 | 2.0 | 4.0 | triglyme | 24 | 53 |
| 14 | 2.0 | 3.0 | 4.0 | triglyme | 24 | 64 |
| 15[c] | 3.0 | 2.0 | 4.0 | triglyme | 24 | 93 (89) |
| 16[d] | 3.0 | 2.0 | 4.0 | triglyme | 24 | 70 |
| 17[e] | 3.0 | 2.0 | 4.0 | triglyme | 24 | 44 |
| 18[f] | 3.0 | 2.0 | 4.0 | triglyme | 24 | 13 |

[a]Reactions were conducted with the indicated reagents under the indicated conditions: **1a** (17.2 mg, 0.1 mmol), **3a**, KO$^t$Bu, and the solvent (0.5 mL) reacted at room temperature for the indicated hours.
[b]Determined using $^{19}$F and $^1$H nuclear magnetic resonance (NMR) spectroscopy with 3-fluoropyridine as an internal standard. The number in parentheses refers to the isolated yield, and those in the columns titled **3a**, **Si–B**, and KO$^t$Bu refer to molar equivalents.
[c]Reaction was performed at the 0.2-mmol scale.
[d]$^t$BuMe$_2$SiBpin was used instead of Et$_3$SiBpin.
[e]PhMe$_2$SiBpin was used instead of Et$_3$SiBpin.
[f](Me$_3$Si)$_3$SiBpin was used instead of Et$_3$SiBpin. diglyme: diethylene glycol dimethyl ether. 18-crown-6: 18-crown-6-ether. triglyme: triethylene glycol dimethyl ether. **Si–B**: silylboronates, such as Et$_3$SiBpin, $^t$BuMe$_2$SiBpin, PhMe$_2$SiBpin, (Me$_3$Si)$_3$SiBpin.

substrate **1c**, reacted efficiently with **3a** under the optimal conditions to generate the corresponding cross-coupling amination products (**4aa**–**4da**) in high yields (80–89%). The use of non-substituted (**1e**) and *para*-substituted fluorobenzenes with electron-donating (Me, **1f**; MeO, **1g**) or electron-withdrawing groups (CF$_3$, **1h**) also successfully yielded the corresponding products (**4ea**, 79%; **4fa**, 74%; **4ga**, 66%; **4ha**, 82%) via defluoroamination with **3a**. The chemoselectivity of this defluoroamination process was achieved successfully when other halide-substituted aryl fluorides (Cl, **1i**; Br, **1j**) were employed (**4ia**, 51%; **4ja**, 39%). Furthermore, the fluoroarenes **1k**–**1t**, which contained π-extended moieties with various electronic properties, were efficiently converted to the corresponding cross-coupling amination products **4ka**–**4ta** in good-to-high yields (51–86%) and were almost independent of the attached functional group. Remarkably, the excellent chemoselectivity of this process was revealed by the tolerance toward the reaction conditions of functional groups such as ethers (OMe, **1m**; OBn, **1n**) and Cl (**1o**), Br (**1p**), CN (**1q**), and CF$_3$ groups (**1r**, **1s**, and **1t**), which may be cleaved via C–F bond activation; for example, 4-(naphthalen-1-yl)phenyl- (**4ka**: 77%), 4-methylphenyl- (**4la**: 86%), and ether-containing biphenyl products (**4ma**: 83%; **4na**: 61%) and biphenyl products containing electron-withdrawing groups (**4oa**: 77%; **4pa**: 59%; **4qa**: 51%; **4ra**: 84%; **4sa**: 82%; **4ta**: 54%). The aryl fluoride-containing benzo[1,3]dioxole, **1u**, was also converted to the defluoroamination product **4ua** and obtained a yield of 81% without cleaving the C–O bonds. Additionally, *N*-containing heteroaromatic fluorides (**1v**, **1w**, **1x**, **1y**, **1z**, and **1aa**) were successfully defluoroaminated using **3a** under the same conditions in higher yields (≤97%).

Pyridine derivatives (**4va**: 93%; **4wa**: 91%; **4xa**: 94%; **4ya**: 94%) and a 1*H*-pyrrole derivative (**4za**: 97%) were obtained via the cross-coupling reactions. Indole- (**1aa** and **1ab**) and benzofuran-containing (**1ac**) aryl fluorides were also functionalized well even though they contained several reactive aryl C(sp$^2$)–H bonds, selectively yielding defluorinative amination products (**4aaa**: 85%; **4aba**: 96%; **4aca**: 90%) via C–F bond cleavage without forming the corresponding C–H bond-activated byproducts. All results clearly demonstrate the remarkable functional group tolerance of these silylboronate-mediated cross-coupling amination reactions of aryl fluorides and amines.

Next, substituted *N*-methylanilines **3** were examined via coupling with **1a** under standard conditions. Methyl substituents at the *para*- (**3b**), *meta*- (**3c**), or *ortho*- (**3d** and **3e**) positions of *N*-methylaniline were evaluated via reaction with **1a**. High yields of coupling products (**4ab**: 87%; **4ac**: 82%) were obtained using **3b** or **3c** and **1a**, while the yields of the sterically hindered products (**4ad**: 31%; **4ae**: 32%) were low. *N*-Methylanilines bearing electron-donating (4-OMe: **3f**, 1,3-dioxole: **3g**) or electron-withdrawing (4-OCF$_3$: **3h**; 4-Cl: **3i**; 3-Cl: **3j**; 4-Br: **3k**) groups underwent defluoroamination to afford the desired products in good yields (**4af**: 61%; **4ag**: 67%; **4ah**: 49%; **4ai**: 60%; **4aj**: 54%; **4ak**: 33%).

Because broad ranges of **1** and **3** are applicable in this coupling reaction, we demonstrated the further scope of the defluoroamination using various combinations of **1** and **3**. Phenyl (**1b**, **1c**), naphthyl (**1d**), electron-donating 4-OMe (**1g**), and electron-withdrawing 4-CF$_3$ (**1r**)-substituted fluorobenzenes were coupled with various *N*-methylanilines (**3c**, **3d**, **3h**) to generate the desired amines **4** in good yields (**4bc**: 76%; **4dh**: 54%; **4gc**: 68%; **4hh**: 45%), including the sterically

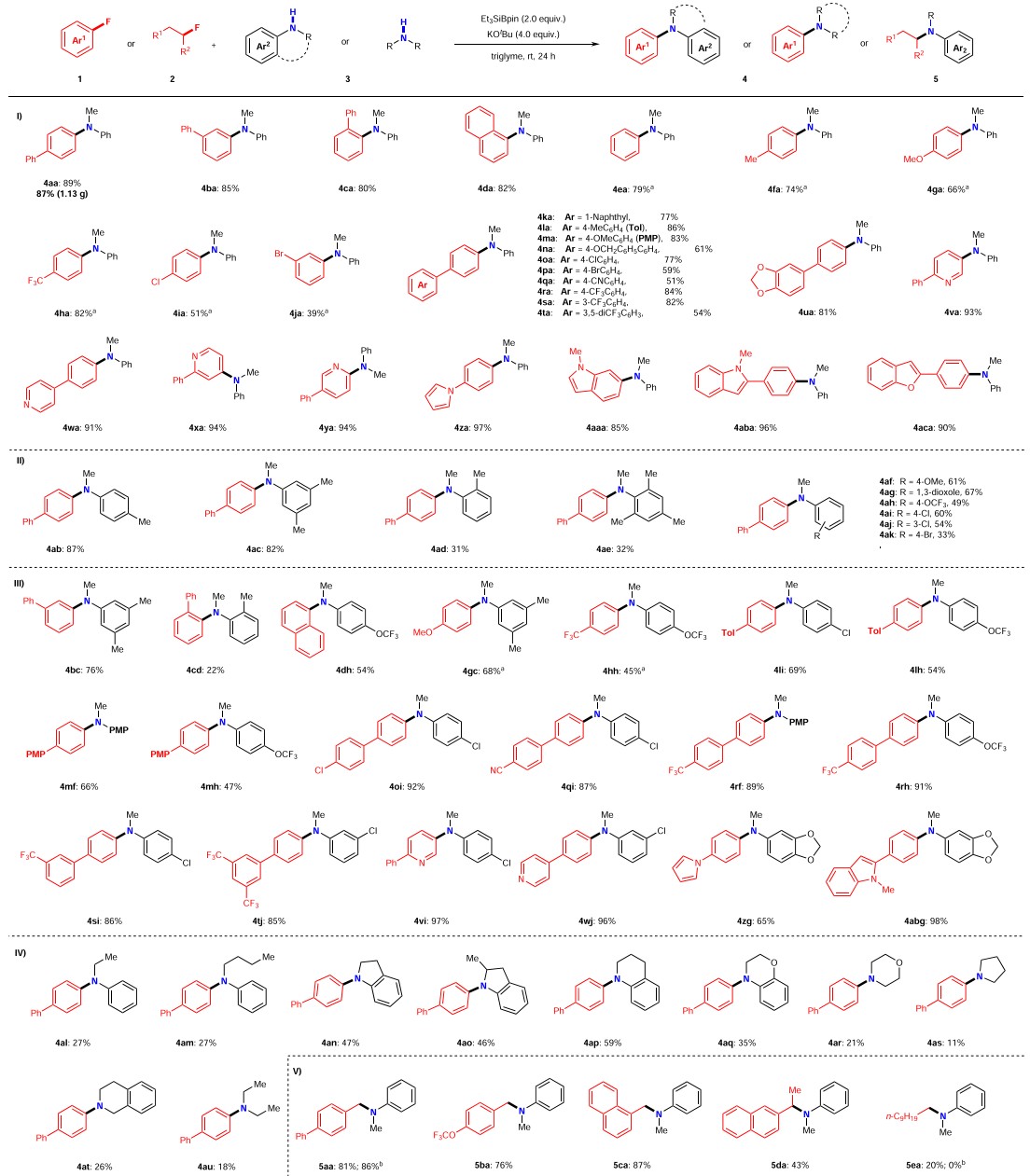

**Fig. 2 | Substrate scope of 1, 2, and 3.** Unless otherwise noted, all reactions were conducted using **1** or **2** (0.2 mmol), **3** (3.0 equiv.), Et₃SiBpin (96.8 mg, 2.0 equiv.), KOtBu (89.6 mg, 4.0 equiv.), and triglyme (1.0 mL) at room temperature for 24 h, with isolated yields shown. [a]Reaction performed using 0.4 mmol of **1**. [b]Reaction performed without Et₃SiBpin, and yield was determined by ¹H NMR.

unfavorable product (**4cd**: 22%). In addition, the use of biphenyl fluorides **1** with electron-donating (**1l**, **1m**), electron-withdrawing (**1q**, **1r**, **1s**, **1t**), or chloride (**1o**) functional groups efficiently generated the corresponding products **4** via reaction with anilines **3** in good yields (**4li**: 69%; **4lh**: 54%; **4mf**: 66%; **4mh**: 47%; **4oi**: 92%; **4qi**: 87%; **4rf**: 89%; **4rh**: 91%; **4si**: 86%; **4tj**: 85%). The reactions of N-heterocycle-containing aryl fluorides (**1v, 1w, 1z, 1ab**) with chloro- (**3i, 3j**) or benzo[1,3]dioxole anilines (**3g**) resulted in good-to-high yields (**4vi**: 97%; **4wj**: 96%; **4zg**: 65%; **4abg**: 98%).

To understand the limitations of this defluorinative coupling with aryl fluorides **1** in terms of secondary amines, several representative secondary amines, such as N-alkylanilines (**3l, 3m**), cyclic anilines (**3n–3q**), and cyclic and acyclic alkylamines (**3r–3u**) were reacted with **1a** under standard conditions. Remarkably, in all cases, the defluoroamination with **1a** generated the desired products, while the yields

varied depending on the type of amine (Fig. 2, IV). The use of N-ethyl- (**3l**) or N-butylaniline (**3m**) afforded the corresponding products in low yields (**4al**: 27%; **4am**: 27%) via coupling with **1a**. Conversely, cyclic N-alkylanilines reacted more favorably with **1a** to produce the corresponding products in moderate yields (**4an**: 47%; **4ao**: 46%; **4ap**: 59%; **4aq**: 35%). Therefore, the steric hindrance around the N center is strongly related to the reactivity, and the bulkier anilines generally afford lower yields. Notably, this coupling reaction can be extended to non-aryl cyclic and acyclic dialkyl secondary amines, such as morpholine (**3r**), pyrrolidine (**3s**), 1,2,3,4-tetrahydroisoquinoline (**3t**), and diethylamine (**3u**), in generating the corresponding products (**4ar**: 21%; **4as**: 11%; **4at**: 26%; **4au**: 18%). However, in these cases, **1a** was partially consumed. Additionally, byproducts, mainly the defluorosilylation product, PhC₆H₄-SiEt₃ (**6**), were detected[42]. The low yields and the formation of byproducts are possibly attributed to the low

**Fig. 3 | Synthetic applications. a–c** Late-stage cross-coupling aminations of pharmaceutically attractive molecules. **d–g** Late-stage syntheses of pharmaceutically attractive molecules containing deuterated *N*-methyl groups.

reactivities of the reactants and the steric hindrance of the aniline moieties.

Finally, we attempted the defluorinative aminations of alkyl fluorides **2** containing C(sp³)−F bonds using **3a** under optimal reaction conditions (Fig. 2, V). Primary benzyl fluorides (**2a–2c**) efficiently underwent defluoroamination with **3a** to generate the desired coupling products in the yields of ≤87% (**5aa**: 81%; **5ba**: 76%; **5ca**: 87%). Notably, the C(sp³)−F bond in **2b** was cleaved without affecting the OCF₃ substituent. Further, using the secondary benzyl fluoride 2-(1-fluoroethyl)naphthalene (**2d**) yielded the corresponding defluoroamination product **5da** in a yield of 43%. In contrast, **5ea** was obtained in a yield of 20% using 1-fluorodecane (**1e**) and **3a** under identical standard reaction conditions. Thus, the reactions of non-activated alkyl fluorides are limited using this method, although the yields may be improved via the extensive optimization of the reaction conditions. Interestingly, while benzyl fluoride **2a** gave the amination product **5aa** in 86% yield even in the absence of Et₃SiBpin, 1-fluorodecane (**1e**) remained unreacted.

## Application of silylboronate-mediated defluorinative coupling reaction

To highlight the synthetic applications of this silylboronate-mediated defluorinative coupling reaction, we examined the functionalization of several drug derivatives with fluoroarene moieties (Fig. 3). The (±)-α-tocopherol-derived fluoroarene **1ad** underwent the coupling reaction with **3a** to generate the (±)-α-tocopherol derivative **4ada** in a yield of 83%. Moreover, the bioactive motif (-)-menthol-derived *N*-methylaniline **3v** was successfully functionalized using this reaction with **1a** or the fluoro-containing estrone derivative **1ae** to generate **4av** and **4aev** in yields of 75% and 62%, respectively. Furthermore, we extended the protocol to the late-stage syntheses of deuterated *N*-alkyl

pharmaceuticals. Because >50% of the best-selling drugs contain *N*-alkyl groups[49], the development of deuterated *N*-alkyl pharmaceuticals is gaining considerable attention. Isotope labeling is generally critical in medicinal chemistry as the C–D bonds are more stable than C–H bonds[50]. Therefore, incorporating deuterated *N*-methyl (*N*-CD₃) moieties into pharmaceuticals should improve the pharmacodynamic properties[51–56]. Considering this aspect, several representative fluoro-containing derivatives of bioactive molecules, **1ad–1ag**, were reacted with deuterated *N*-methylaniline *d³*-**3a** under standard conditions. **1ae** and *d³*-**3a** were initially used in the presence of Et₃SiBpin and KO*t*Bu in triglyme, generating the estrone derivative *d³*-**4aea** in a 77% yield. The reaction with (-)-menthol-derived fluorobenzene **1af** also proceeded well under identical conditions to afford the deuterated product *d³*-**4afa** in a yield of 88%. The deuterated (*R*)-naproxen derivative *d³*-**4aga** was also synthesized in a 70% yield using the fluoropropanoate **1ag**. Moreover, **1ad** was again investigated via reaction with *d³*-**3a** under standard conditions to yield the corresponding deuterated (±)-α-tocopherol derivative *d³*-**4ada** (79% yield).

## Reaction mechanism

Several control experiments were conducted to gain insight into the reaction mechanism (Fig. 4, left). First, the silylated compound **6** was used in the coupling reaction with **3a** under optimal reaction conditions. However, the amination product **4aa** was not detected (Fig. 4a); thus, **6** did not participate in the reaction. We then examined the defluoroamination of **1a** with **3a** in the presence of (2,2,6,6-tetramethylpiperidin-1-yl)oxyl (TEMPO, Fig. 4b). Although **4aa** was obtained in a yield of 89% under standard conditions, the yields considerably decreased when the amount of TEMPO increased: 40% (1.0 equiv. of TEMPO), 26% (2.0 equiv. of TEMPO), and 9% (4.0 equiv. of TEMPO). The reaction of benzyl fluoride **2a** was also inhibited by the

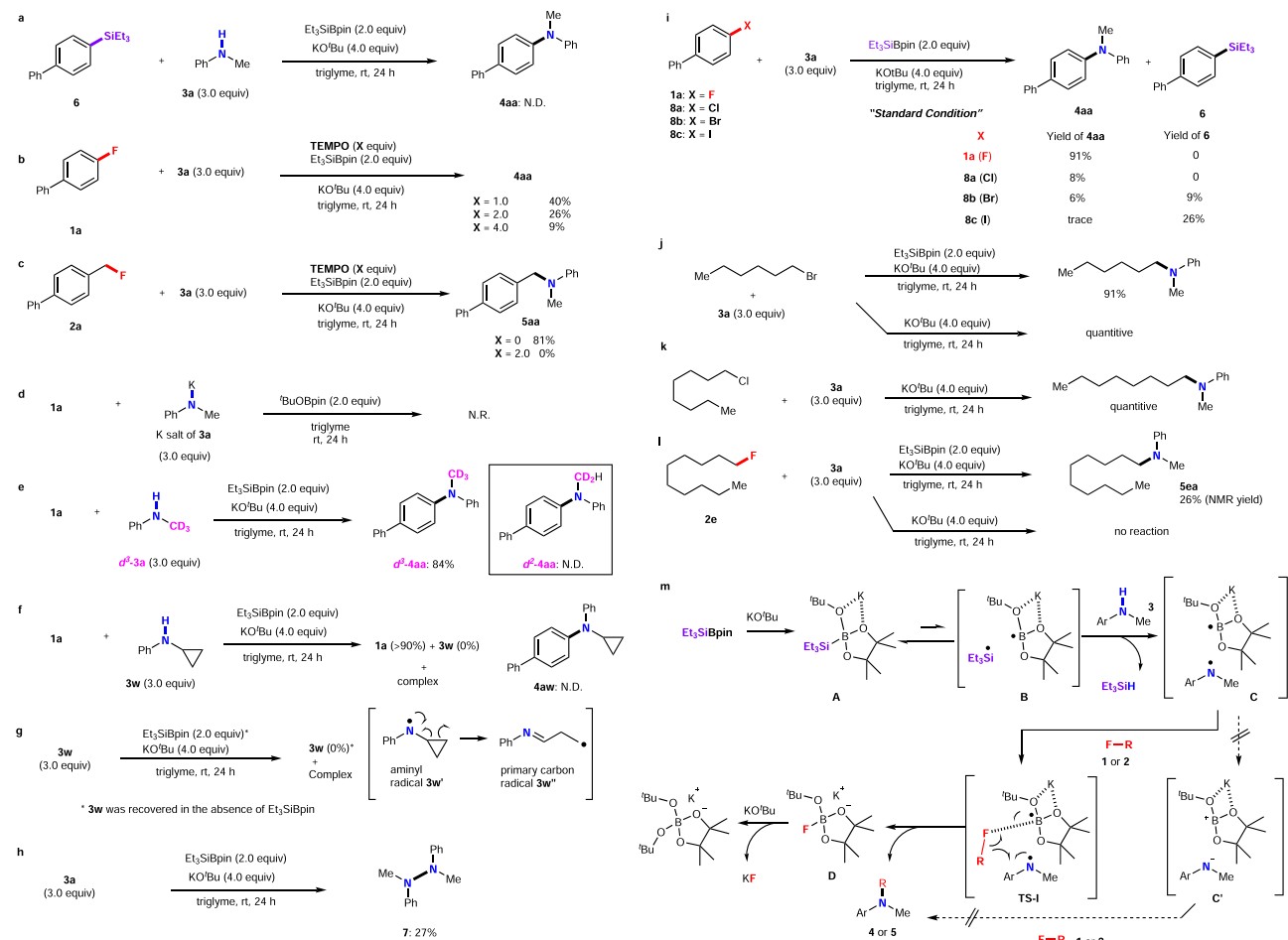

**Fig. 4 | Mechanistic studies. a** Stepwise attempted reaction. **b, c** Effect of TEMPO on the silylboronate-mediated coupling reaction. **d** Reaction of **1a** with the potassium salt of **3a** (PhNKMe) in the presence of ʹBuOBpin. **e** Reaction between **1a** and the deuterated **3a** (**d³-3a**). **f, g** Radical clock experiments. **h** Evidence suggesting the generation of an *N*-methylanilino radical. **i** Chemoselectivites of organic halides Ar–X. **j–l** Different reactivities of alkyl halides under the standard conditions in the presence and absence of Et₃SiBipin. **m** Proposed reaction mechanism.

addition of TEMPO (2.0 equiv. Figure 4c). The reaction of **1** with the freshly prepared potassium salt of **3a** in the presence of ʹBuOBpin[57] was attempted; however, no reaction occurred (Fig. 4d). Thus, the process should not involve the formation of potassium anilide. Furthermore, **d³-3a** was employed under optimal reaction conditions, and the corresponding defluoroamination product **d³-4aa** was isolated in an 84% yield (Fig. 4e), whereas **d²-4aa** was not obtained. Therefore, the *N*-methyl moiety is not involved in the process. In addition, a radical clock experiment with **1a** and *N*-cyclopropylaniline (**3w**) was conducted. **1a** was almost recovered (about 90%), whereas **3w** disappeared to give a complex mixture (Fig. 4f). Furthermore, the treatment of only **3w** under standard conditions resulted in the same complex mixture, whereas **3w** was recovered under the same treatment in the absence of Et₃SiBPin (Fig. 4g). These facts are consistent with the ring-opening of the aminyl radical **3w'** into the primary carbon radical **3w''** under Et₃SiBpin/ʹBuOK conditions[58–60]. The ring-opening primary carbon radical **3w''** is unstable towards decomposition (Fig. 4f-g). When the reaction of **3a** was conducted without **1a**, hydrazine **7** was formed in an isolated yield of 27% (Fig. 4h) due to the dimerization of the *N*-methylanilino radical PhN•Me. These control experiments indicate that the defluorinative C–N cross-coupling proceeds via a radical pathway. The uniqueness of this reaction using aryl fluorides **1** rather than the conventional aryl halides Ar–X **8a–c** (X = Cl, Br, or I) in the silylboronate-mediated cross-coupling was revealed via comparison studies (Fig. 4i). Under identical conditions, 4-chlorobiphenyl (**8a**) was converted to the desired coupling product **4aa**; however, the yield of

8% was inadequate. In contrast, the use of bromo- or iodo-substituted biphenyl (Br: **8b**; I: **8c**) generated a mixture of amination product **4aa** and silylation product **6**: **4aa** (Br: 6% and I: trace) and 4-biphenylyl(-trimethyl)silane **6** (Br: 9% and I: 26%), respectively. Thus, the chemoselectivity of this cross-coupling reaction toward the C–F bond over C–Cl/Br/I bonds is excellent.

Next, we compared the reactivities of alkyl halides under the standard conditions in the presence and absence of Et₃SiBpin. Alkyl bromide reacted with *N*-Me-aniline **3a** under standard conditions, affording the desired amination product in 91% yield (Fig. 4j). However, even in the absence of Et₃SiBpin, the amination product formed quantitatively (Fig. 4j). Alkyl chloride also gave the desired amination product in high yield without Et₃SiBpin (Fig. 4k). Alkyl fluoride **2e** gave **5ea** under standard conditions, but no reaction occurred in the absence of Et₃SiBpin (Fig. 4l). Notably, the alkyl bromides and chlorides, and not alkyl fluorides, react easily with the nucleophilic species by a common Sₙ2 reaction[61]. These results indicate the apparent difference between the C–F and C–X (Br, Cl) bonds and demonstrate that C–F bond cleavage is indeed challenging.

Based on our results and those reported in previous studies, we propose a single-electron-transfer/radical-mediated defluorinative amination mechanism using frustrated radical pair chemistry[62–67] (Fig. 4m). Initially, Et₃SiBpin reacts with KOʹBu to form an intermediate, **A**. The formation of this intermediate was previously proposed by Avasare et al. based on density functional theory calculations[68]. **A** was also identified via ¹¹B and ²⁹Si nuclear magnetic resonance (NMR)

studies[42,43]. The increase in the steric hindrance of the alkyl moieties of $R_3SiBpin$ decreased the conversion and the yields of the reactions (Table 1, entries 16–18). This observation indicates that the initial step of the nucleophilic addition of $t$-butoxy anion ($^-O^tBu$) to $R_3SiBpin$ highly depends on the steric factor and is independent of the radical stability of the $R_3Si$ species generated in the next step. These results also agree with the solvent effect (Table 1, entries 4–6). It is known that the potassium cation is capsuled by glymes and by 18-crown-6, which results in the generation of naked, reactive $^-O^tBu$[69–73]. The naked $^-O^tBu$ anion should accelerate the generation of intermediate **A**. Next, owing to a single electron reductant property of $^-O^tBu$[74,75], intermediate **A** splits into a sterically demanding frustrated radical pair of **B**, which comprises the triethylsilyl radical ($•SiEt_3$) and a boron-radical species (B•), via the homolytic cleavage of the Si−B bond. Hydrogen abstraction from $N$-methylaniline **3** by $•SiEt_3$ in **B** yields a frustrated radical pair **C** consisting of a $N$-methylanilino radical (Ar-$N$•-Me) and the boron-radical species, accompanied by the formation of $HSiEt_3$ (detected using gas chromatography-mass spectrometry). The generation of the $N$-methylanilino radical is supported by the low yields of **4aa** obtained under optimal reaction conditions in the presence of increased equivalents of TEMPO. Subsequently, the frustrated radical pair **C** attracts organic fluoride **1** (or **2**) by preferable interaction between the F atom and B center to afford **TS-I**. Since the C−F bond of **1** (or **2**) is activated by the interaction of the B atom in **TS-I**, $N$-methylanilino radical selectively attacks at the carbon center of the C−F bond, as shown in **TS-I**. Finally, the desired cross-coupling product **4** or **5** is obtained with C−N bond formation, accompanied by the release of **D** ([Bpin($O^tBu$)F]K), which promptly reacts with $KO^tBu$ to provide a stable ([Bpin($O^tBu$)$_2$] species (detected using $^{11}B$ NMR spectroscopy) and KF (detected using $^{19}F$ NMR spectroscopy) (see the Supplementary Information). The fluorine-selective reaction over the other halogens could be explained based on bond dissociation energy (BDE)[76]. The C−F bond in Ph−F (BDE: 125.6 kcal/mol) is more stable than the C–Br (BDE: 80.4 kcal/mol) and C–Cl (95.5 kcal/mol) bonds in Ph−X. However, the B–F (boron-fluorine) bond is the strongest among the other B–X bonds (X = F, Cl, Br) in terms of the reported (B(O)-X; X–F: 163.0 kcal/mol), X–Cl: 104.6 kcal/mol; X–Br: 86.7 kcal/mol)[61] and predicted values of the corresponding intermediate structures (PinB⁻(X)$O^tBu$; X = F: 105.2 kcal/mol; X = Cl: 91.9 kcal/mol; X = Br: 83.9 kcal/mol)[77]. Therefore, the B atom of the intermediate selectively approaches the F atom over the Cl or Br atoms on the substrates, resulting in the preferential attack of the amino radical at the carbon attached to the F atom over other halogen-attached carbons. The preferable interactions between the F and B centers are essential for the cleavage of the C−F bond instead of those between the B centers and other halogens. Another mechanistic pathway based on the nucleophilic $S_NAr$ reaction of organic fluoride **1** or **2** with the anilino anion in the frustrated ion pair **C′** was ruled out by the control experiments shown in Fig. 4d.

In conclusion, we developed the first silylboronate-mediated radical coupling of organic fluorides with secondary amines via inert C−F bond activation at room temperature. A wide variety of secondary acyclic and cyclic $N$-alkylanilines and dialkylamines reacted with different organic fluorides to generate structurally diverse aromatic tertiary amines in moderate-to-excellent yields under very mild conditions. The most significant feature of this protocol is the production of electrically opposing scaffolds, particularly tolerating reactants containing OR-, Cl-, CN-, or $CF_3$ functional groups. Additionally, this method avoids the use of transition metals and specialized ligands. While the Buchwald−Hartwig amination is undoubtedly a powerful strategy to access several secondary/tertiary amines by using aryl halides (I, Br, and Cl) and the Pd catalyst system, the use of aryl fluorides is not practical in this reaction. However, as the utilization of fluorine-containing drugs in pharmaceutical and agrochemical industries has recently increased[8,9], the present C−F bond transformation strategy is desirable for preparing new drug candidates. In fact, the late-stage syntheses of $N$-alkyl pharmaceuticals, including deuterated analogs, were effectively achieved. Due to the very mild reaction conditions, ease of execution, and wide substrate scope of this protocol, in addition to the structurally derivatizable complex pharmaceuticals produced, this silylboronate-mediated C−F and N−H bond coupling should be used in organic syntheses, pharmaceuticals, and agrochemicals. Further studies regarding this defluorinative functionalization are currently underway.

## Methods

### General procedure of the silylboronate-mediated cross-coupling reactions of organic fluorides and secondary amines

In an $N_2$-filled glovebox, organic fluorides **1** or **2** (0.2 mmol, 1.0 equiv.), $KO^tBu$ (89.6 mg, 0.8 mmol, 4.0 equiv.), dry triglyme (1.0 mL), secondary amines **3** (0.6 mmol, 3.0 equiv.), and $Et_3SiBpin$ (0.4 mmol, 2.0 equiv.) were sequentially added to a flame-dried screw-capped test tube. The tube was then sealed and removed from the glovebox, and the solution was stirred at room temperature for 24 h. The mixture was diluted (5 mL) and then extracted using diethyl ether and water, washed with brine, dried over $Na_2SO_4$, and concentrated under a vacuum. Subsequently, 3-fluoropyridine (8.6 µL, 0.1 mmol) was added as an internal standard before NMR spectroscopy. The mixture was then concentrated again to yield the crude product, which was purified via column chromatography on silica gel to yield the corresponding products **4** or **5**.

## Data availability

The data that support the findings of this study are available within the article and the Supplementary Information. Details about materials and methods, experimental procedures, characterization data, mechanistic studies, and NMR spectra are available in the Supplementary Information. All relevant data are also available from the authors.

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

## Acknowledgements

This study was supported by the CREST program of the Japan Science and Technology Agency entitled "Precise Material Science for Degradation and Stability" (Grant number JPMJCR21L1).

## Author contributions

J.Z. and Z.Z. performed the experiments, analyzed the data, and then discussed the results with N.S. J.Z. and N.S. wrote the paper, and N.S. supervised the project. All authors contributed to the paper and approved the final version.

## Competing interests

The authors declare no competing interests.
