## [Peer Review File · Nature Communications]

REVIEWER COMMENTS

Reviewer #1 (Remarks to the Author):

This paper reports that tertiary amines can be synthesized by the formation of carbon-nitrogen bonds under reaction conditions combining stoichiometric amounts of silylboronates, Et₃SiBpin, and KOtBu as a base, from with secondary amines via chemoselective cleavage of sp² and sp³ carbon-fluorine bonds of aryl or alkyl fluorides. The formation of carbon-nitrogen bonds is a key transformation reaction in synthetic organic chemistry, and the development of synthetic methodologies for N-alkyl diarylamines is an extremely important research topic related to the development of natural product organic compounds, biologically active molecules, and functional materials. In particular, it is of great interest that in the present study, in addition to being able to activate the chemically stable carbon-fluorine bond without the use of a transition metal catalyst, it also reacts prior to other halogens that are more chemically active.

However, for the following reasons, the reviewer concluded that the paper could not be accepted in its current state.

1) The first point to evaluate in this paper is how compatible the substrates are compared to other reported studies without transition metal catalysts. In comparison with the cited references 31-33, similar amination reactions of C-F bonds have been reported when using strong bases such as KOH and nBuLi, and it is known that functional groups such as -OMe, -CN, and -CF₃ are also tolerated in these reactions. Therefore, the authors' claim that "The potentially cleavable C-O bonds of ethers, and C-CN bond are unaffected. Most significantly, the C(sp³)-F bonds of the CF₃ and OCF₃ groups are unaffected." is not novel. Thus, specific and detailed mention should be made of the functional group tolerance achieved only in the present reaction. Otherwise, the authors' statement that "All results clearly demonstrate the remarkable functional group tolerance of these silylboronate-mediated cross-coupling amination reactions of aryl fluorides and amines." would be weak.

2) Regarding the applicability of aryl fluorides, the authors state that "Although the defluorinative aminations of aryl fluorides are reported, the aminations of electron-rich aryl fluorides are rare.^{34,35}" However, this is inconsistent with the fact that many examples of reactions using para-methylphenyl and para-methoxyphenyl groups are presented in both papers of 34 and 35.

3) Third, in terms of the chemoselective activation of the C-F bond, it has recently been reported that the (sp²)carbon-fluorine bond can be activated preferentially over the (sp²)carbon-bromine bond using organophotoredox catalysts, as shown in reference 34. This paper lacks impact considering that it is a stoichiometric reaction.

4) It is noteworthy that this paper shows that many N-alkyl diarylamines can be synthesized as new compounds, but since these compounds have different aryl groups, they have not been addressed as

synthetic targets, but can be synthesized using other synthetic methods, e.g., Buchwald-Hartwig amination.

Meanwhile, the reviewer would like to address the following questions and suggestions regarding the content of this paper. The authors will use them to improve this paper for resubmission.

- 1) In entry 13 of Table 1, the authors emphasize reproducibility at the 0.2 mmol scale, but it is necessary to provide at least a gram-scale experiment to demonstrate the reproducibility.
- 2) The authors need to discuss the reasons for the generally low yields of the products in Table 2 IV) (e.g., presence of unreacted substrates or side reactions).
- 3) In the reaction mechanism shown in Fig. 3g, it should be discussed why carbon-fluorine bond reacts with aminino radical preferentially over other halogens, showing bond dissociation energies (BDE).
- 4) Although comparable results have been obtained for substituents on silicon with the PhMe₂Si group as well as Et₃Si, it would be desirable for the authors to address the stability/reactivity of the silyl radicals that are initially generated. In addition, it is assumed that the rate at which the aminino radical is generated by one-electron transfer (hydrogen abstraction) from the initially generated silyl radical is quite fast. Therefore, it would be even better if a discussion of the stability of each chemical species could be added.
- 5) When substrates with (sp³)carbon-bromine bonds are used, what results are obtained?
- 6) The role of the solvent should be discussed, given that comparable results have been obtained using 18-crown-6 as a solvent.

Minor comments:

- 1) Page 6, line 115: "halide (1m)" should be corrected to "chloride (1m)".
- 2) What results are given when allyl fluoride or cinnamyl fluoride is used as a substrate with (sp³)carbon-fluorine bond?
- 3) References 35 and 41 are from "Chem.-Eur. J."
- 4) Page 9, lines 116-117, lines 173-174: "Hence, the cross-coupling reaction should involve radical species." and "These control experiments indicate that the defluorinative C–N cross-coupling proceeds via a radical pathway." is a repetition of the same principle.
- 5) Page 10, line 184: "carbosilylation" should be corrected to "defluorinative amination".
- 6) Page 10, line 200: "Conclusion" instead of "Discussion".

Regarding Supplementary Information,

1) There are no page numbers.

2) There are many words in the text that should not begin with a capital letter. For example, "Dioxane, DME, "D"iethylene glycol dimethyl ether ("D"iglyme), "T"riethylene glycol dimethyl ether ("T"riglyme), "C"yclopentyl methyl ether (CPME) and THF were dried and distilled before use." and "In a flame-dried flask was charged with "A"nilines (10 mmol), "M"ethyl iodide (12 mmol), "P"otassium carbonate (13 mmol), and N,N-"D"imethylformamide (DMF, 25 mL)."

3) Tables 2 and 3: "arylfluoride" should be spaced with "aryl fluoride".

4) For "Silylboronate Et₃SiB"Pin were prepared according to the procedures previously 1 and 4" should be the original paper, not the paper by the authors themselves.

Reviewer #3 (Remarks to the Author):

This paper by Shibata and co-workers deals with the silylboronate-mediated selective defluorinative cross-coupling of organic fluorides with secondary amines via a transition-metal-free strategy and is strongly based on some previous reports. In particular, the authors recently published a paper (Org. Lett. 2022, 24, 508) whose title (Etherification of Fluoroarenes with Alkoxyboronic Acid Pinacol Esters via C–F Bond Cleavage) clearly indicates the close relation with the present paper. In fact, the only essential difference is the use of amines for the generation of N-radical species, instead of an alkoxy radical derived from RO-Bpin in the mentioned communication. What is indeed very surprising is the fact that the authors omitted the citation of this paper. Therefore, the novelty in this paper lies just in the development of an extension of this known process, which is indeed a valuable contribution, but is far from meeting the strict novelty criteria for publication in a leading journal such as Nature Communications.

Prof Shibata et al have developed a novel method for the cross coupling of aryl fluorides with secondary amines using super-stoichiometric amounts of reagents Et_3SiBPi , KOtBu and R_2NH . The paper is presented in a logical manner and the prose is of a high degree of scholarship. The scope of the reaction includes mostly para-substituted aryl fluorides with arylamines, although the authors have extended the reaction to a few aliphatic amines as well as ortho- and meta-substituted aryl fluorides, albeit in reduced yields. Nevertheless, the authors have developed an impressive scope of reactivity involving multiple functional groups, EWGs/EDGs, and have applied their reaction to the synthesis of potentially high-value pharmaceutical targets. All newly synthesized molecules are well characterized.

However, the proposed mechanism is likely incorrect, and a much simpler alternative is available that is also consistent with their mechanistic findings. The authors propose a SET event to produce intermediate B (Fig 3), which is a dianionic boryl, as well as $\text{Et}_3\text{Si}\cdot$ and $\text{tBuO}\cdot$. There are a number of things troubling with this – surely the two radical components would immediately recombine to produce stable Et_3SiOtBu , even at least to some degree if they were present in the reaction? The authors however state that only Et_3SiH is detected in the reaction mixture. Furthermore, intermediate B looks atrociously unstable, and to my knowledge a dianionic boryl $[(\text{RO})_3\text{B}]^{2-}$ has no precedence in low-valent boron chemistry, and similar species can only be stabilized by the presence of strong pi-acceptors, such as cyanide. There seems to be a typo at intermediate C where an extra hydrogen atom has appeared. While the intermediacy of an amido radical is indicated from the control experiments, the 4-component radical cascade involving double radical transfer from intermediate B seems very dubious. There is a much simpler alternative:

This seems to be fully consistent with the observed results and the structures proposed have a good deal of precedence in Frustrated radical pair chemistry. This would also be easy to test – combine KNR_2 with $\text{BPin}(\text{OtBu})$ and see if this performs nucleophilic substitution on aryl fluorides.

Also, the mechanism in Fig 3 is drawn as a circle, which usually indicates catalysis. This is not catalysis - this reaction is super-stoichiometric in every reagent. To draw it as a circle seems misleading.

Once these corrections are made I would find the article suitable for publication. Other than the mechanistic work, the conclusions of the paper are well substantiated, the method described is simple, novel, seems to be fairly generally applicable, and is a useful addition to the chemical toolbox to make C-N bonds.

RESPONSE TO REVIEWERS:

We thank you for your thoughtful suggestions and insights, which have enriched the manuscript and produced a better and more balanced account of the research.

Reviewer #1 (Remarks to the Author):

This paper reports that tertiary amines can be synthesized by the formation of carbon-nitrogen bonds under reaction conditions combining stoichiometric amounts of silylboronates, Et_3SiBpin , and $\text{KO}t\text{Bu}$ as a base, from with secondary amines via chemoselective cleavage of sp^2 and sp^3 carbon-fluorine bonds of aryl or alkyl fluorides. The formation of carbon-nitrogen bonds is a key transformation reaction in synthetic organic chemistry, and the development of synthetic methodologies for *N*-alkyl diarylamines is an extremely important research topic related to the development of natural product organic compounds, biologically active molecules, and functional materials. In particular, it is of great interest that in the present study, in addition to being able to activate the chemically stable carbon-fluorine bond without the use of a transition metal catalyst, it also reacts prior to other halogens that are more chemically active.

However, for the following reasons, the reviewer concluded that the paper could not be accepted in its current state.

1) The first point to evaluate in this paper is how compatible the substrates are compared to other reported studies without transition metal catalysts. In comparison with the cited references 31-33, similar amination reactions of C-F bonds have been reported when using strong bases such as KOH and *n*- BuLi , and it is known that functional groups such as $-\text{OMe}$, $-\text{CN}$, and $-\text{CF}_3$ are also tolerated in these reactions. Therefore, the authors' claim that "The potentially cleavable C-O bonds of ethers, and C-CN bond are unaffected. Most significantly, the $\text{C}(\text{sp}^3)\text{-F}$ bonds of the CF_3 and OCF_3 groups are unaffected." is not novel. Thus, specific and detailed mention should be made of the functional group tolerance achieved only in the present reaction. Otherwise, the authors' statement that "All results clearly demonstrate the remarkable functional group tolerance of these silylboronate-mediated cross-coupling amination reactions of aryl fluorides and amines." would be weak.

Response: Thank you for bringing this to our attention. We understand your concerns. In the reactions reported in references 31–33, the defluoro-amination products are obtained as mixtures of regio-isomers because these reactions proceed via benzyne intermediates. Our method is remarkably different from these reported methods as it does not have such limitations. Thus, we have accordingly revised the text.

2) Regarding the applicability of aryl fluorides, the authors state that "Although the defluorinative aminations of aryl fluorides are reported, the aminations of electron-rich aryl fluorides are rare.^{34,35}" However, this is inconsistent with the fact that many examples of reactions using para-methylphenyl and para-methoxyphenyl groups are presented in both papers of 34 and 35.

Response: The reported methods (34 and 35) **require the OMe groups (EDG) on the aromatic rings to activate the substrates. Our method does not require the activation of the aromatic moieties by either EDGs or EWGs.** Thus, we have rephrased the text in the introduction accordingly.

references 34-35

3) Third, in terms of the chemoselective activation of the C–F bond, it has recently been reported that the (sp²)carbon-fluorine bond can be activated preferentially over the (sp²)carbon-bromine bond using organophotoredox catalysts, as shown in reference 34. This paper lacks impact considering that it is a stoichiometric reaction.

Response: The method described in reference 34 indicates that C–F bond activation preferentially occurs over the (sp²)C–Br bond when an organophotoredox catalyst is used. However, this method **requires an electron-donating OMe group to realize the C–F bond activation.** In contrast, **our method does not require the activation of aromatic moieties using EDGs or EWGs.** To clearly demonstrate the chemoselective activation of the C–F bond over the C–Br and C–Cl bonds, we have added four more examples (**4ia, 4ja, 4pa, 4ak**). The original manuscript indicated examples of the Br- and Cl-containing substrates, **4li, 4oi, 4si, 4vi, and 4wj.**

reference 34

4) It is noteworthy that this paper shows that many *N*-alkyl diarylamines can be synthesized as new compounds, but since these compounds have different aryl groups, they have not been addressed as synthetic targets but can be synthesized using other synthetic methods, e.g., Buchwald-Hartwig amination.

Response: While the Buchwald–Hartwig amination reaction is undoubtedly a powerful strategy to access several secondary/tertiary amines using aryl halides (I, Br and Cl) under the Pd catalyst system, the use of aryl fluorides is not practical in this reaction. However, as the utilization of fluorine-containing molecules in pharmaceutical and agrochemical industries has recently increased, the C–F bond transformation strategy is more and more desirable for preparing new drug candidates using fluorine-containing drugs

and related fluorinated materials. Therefore, our new method that enables C–N bond coupling by the selective cleavage of the C–F bond under very mild conditions, especially without the use of expensive transition metal catalysts or photocatalysts, is necessary for chemists. We have added this explanation in the conclusion section of the revised manuscript.

Meanwhile, the reviewer would like to address the following questions and suggestions regarding the content of this paper. The authors will use them to improve this paper for resubmission.

1) In entry 13 of Table 1, the authors emphasize reproducibility at the 0.2 mmol scale, but it is necessary to provide at least a gram-scale experiment to demonstrate the reproducibility.

Response: Thank you for pointing this out. The details of a gram-scale experiment using 5.0 mmol of **1a** have been added in the results and discussion section on page 4.

2) The authors need to discuss the reasons for the generally low yields of the products in Table 2 IV) (e.g., presence of unreacted substrates or side reactions).

Response: In these attempts, the starting material (**1a**) was partially consumed. Additionally, many by-products, mainly the defluorosilylation product, $\text{PhC}_6\text{H}_4\text{-SiEt}_3$, were detected (as reported in reference 36). These low yields of the reactions may have been caused by the low reactivities of the reactants and presumably because of steric hindrance. We have added this explanation in the revised manuscript.

3) In the reaction mechanism shown in Fig. 3g, it should be discussed why carbon-fluorine bond reacts with aminino radical preferentially over other halogens, showing bond dissociation energies (BDE).

Response: The C–F bond in Ph–F (BDE: 125.6 kcal/mol) is more stable than the C–Br (BDE: 80.4 kcal/mol) and C–Cl (95.5 kcal/mol) bonds in Ph–X. However, the B–F (boron-fluorine) bond is the strongest among the other B–X bonds (X=F, Cl, Br) in terms of the reported (B(O)–X) and predicted values of the corresponding intermediate structures (PinB(X)O^tBu). Therefore, the B atom of the intermediate selectively approaches the F atom over B or Cl atoms on the substrates, resulting in the selective activation of the C–F bond over the C–Br or C–Cl bonds. Then amino radical preferentially attacks the carbon attached to the F atom over other halogen-attached carbons. We have added this explanation in the revised manuscript.

B–X	BDE: B(O)–X reported ¹	BDE: B [–] –X (predicted) ²
-----	--------------------------------------	---

B-F	163.0	105.2
B-Cl	104.6	91.9
B-Br	86.7	83.9

1. Luo, Y.-R. (2007). *Comprehensive Handbook of Chemical Bond Energies* (1st ed.). CRC Press. <https://doi.org/10.1201/9781420007282>

2. Qi Yang, Yao Li, Jin-Dong Yang, Yidi Liu, Long Zhang*, Sanzhong Luo*, Jin-Pei Cheng, Holistic Prediction of pKa in Diverse Solvents Based on Machine Learning Approach. *Angew. Chem. Int. Ed.* 2020, 59, 19282-19291.

4) Although comparable results have been obtained for substituents on silicon with the PhMe₂Si group as well as Et₃Si, it would be desirable for the authors to address the stability/reactivity of the silyl radicals that are initially generated. In addition, it is assumed that the rate at which the anilino radical is generated by one-electron transfer (hydrogen abstraction) from the initially generated silyl radical is quite fast. Therefore, it would be even better if a discussion of the stability of each chemical species could be added.

Response: As per your suggestion, we examined the reaction using several silylboronates with different radical stabilities of the Si radicals under standard conditions. Interestingly, the conversion and yields of the reaction decreased with the increasing radical stability of the R₃Si moiety of silylboronates (which is equal to steric hindrance). As the generation of anilino radicals by one-electron transfer (hydrogen abstraction) from the initially generated silyl radical should be quite fast, the reaction depends on the initial stage of the formation of the R₃SiBpin and ^tBuOK complex which is strongly affected by steric factor of R₃SiBpin. We have added these results in Table 1 and discussed the reaction mechanism at the end of the results and discussion section.

Entry	Si-B	4aa (%) ^b
1	Et ₃ SiBpin	91
2	^t BuMe ₂ SiBpin	70
3	PhMe ₂ SiBpin	44
4	TMS ₃ SiBpin	13

^a Unless otherwise stated, reactions were conducted with **1a** (17.2 mg, 0.1 mmol), **3a** (32 μ L, 3.0 equiv.),

silylboronates (**Si-B**, 2.0 equiv.), KOtBu (45.0 mg, 4.0 equiv.), and triglyme (0.5 mL) at room temperature for 24 h.

^b Determined by ¹⁹F NMR and ¹H NMR spectroscopy using 3-fluoropyridine as an internal standard.

5) When substrates with (sp³)carbon-bromine bonds are used, what results are obtained?

Response: We evaluated the reaction of alkyl bromide with *N*-Me-aniline **3a** under standard conditions, and desired amination product was obtained in 91% yield. However, even without silylboronate, the amination product formed quantitatively. Alkyl chlorides also afforded the desired amination product in high yield without silylboronate. These observations indicate that alkyl bromides and chlorides react with aniline **3a** by a common S_N2 reaction. In contrast, alkyl fluoride **2e** gave **5ea** under standard conditions; however, the reaction did not proceed in the absence of silylboronate. These results indicate the apparent difference between the C–F and C–X (Br, Cl) bonds and demonstrate that C–F bond cleavage is indeed challenging. These results agree with the observations of S_N2 reaction reported in a previous study (D. O'Hagan, *Chem. Soc. Rev.*, 2008, 37, 308). We have added these new results in the revised Supplementary Information to indicate the difference between the C–F and C–X bonds.

D. O'Hagan, *Chem. Soc. Rev.*, 2008, 37, 308.

X	Relative reaction rate
F	1
Cl	71
Br	3500
I	4500

6) The role of the solvent should be discussed, given that comparable results have been obtained using 18-crown-6 as a solvent.

Answer: The effect of solvents on this reaction can be explained as follows. It is known that the potassium cation is capsuled by glymes and also by 18-crown-6, resulting in the generation of naked, reactive *tert*-butoxide (⁻OtBu) anions (for glymes; Saito, T. et al. *Sci. Rep.* **8**, 11501 (2018); Fujihira, Y. et al. *J. Org.*

Chem. **86**, 5883–5893 (2021); Fujihira, Y. et al. *Beilstein J. Org. Chem.* **17**, 431–438 (2021). For **18-Crown-6**; Kleeberg, C. Z. *Anorg. Allg. Chem.* **637**, 1790–1794 (2011); Alunni, S., Baciocchi, E. & Perucci, P. *J. Org. Chem.* **41**, 2636–2638 (1976); Saito, T. et al. *Sci. Rep.* **8**, 11501 (2018). The naked ${}^t\text{BuO}^-$ anion should accelerate the generation of intermediate **A** due to the high reactivity. These observations are in good agreement with the results obtained using sterically demanding R_3SiBpin (i.e., increasing the steric factor of R_3SiBpin , decrease the yield). We have added this explanation in the revised manuscript.

Minor comments:

1) Page 6, line 115: "halide (1m)" should be corrected to "chloride (1m)".

Answer: Thank you for bringing this to our attention. This correction has been made in the manuscript.

2) What results are given when allyl fluoride or cinnamyl fluoride is used as a substrate with (sp^3) carbon-fluorine bond?

Answer: The reactions of allyl fluoride and cinnamyl fluoride under the standard conditions produced complex mixtures, and we detected no desired products. This may have occurred because allyl fluoride and cinnamyl fluoride are too reactive and unstable. These data were added in the revised Supplementary Information.

3) References 35 and 41 are from "Chem.-Eur. J."

Answer: These references have been accordingly revised.

4) Page 9, lines 116-117, lines 173-174: "Hence, the cross-coupling reaction should involve radical species." and "These control experiments indicate that the defluorinative C–N cross-coupling proceeds via a radical pathway." is a repetition of the same principle.

Answer: Thank you for pointing this out. We have rephrased this sentence accordingly.

5) Page 10, line 184: "carbosilylation" should be corrected to "defluorinative amination".

Answer: Thank you for pointing this out. We have made the necessary correction.

6) Page 10, line 200: "Conclusion" instead of "Discussion".

Answer: This section heading has been accordingly revised.

Regarding Supplementary Information,

1) There are no page numbers.

Answer: Thank you for bringing this to our attention. We have added the page numbers.

2) There are many words in the text that should not begin with a capital letter. For example, "Dioxane, DME, "D"iethylene glycol dimethyl ether ("D"iglyme), "T"riethylene glycol dimethyl ether ("T"riglyme), "C"yclopentyl methyl ether (CPME) and THF were dried and distilled before use." and "In a flame-dried flask was charged with "A"nilines (10 mmol), "M"ethyl iodide (12 mmol), "P"otassium carbonate (13 mmol), and N,N-"D"imethylformamide (DMF, 25 mL)."

Answer: Thank you for pointing this out. We have made the necessary corrections.

3) Tables 2 and 3: "arylfluoride" should be spaced with "aryl fluoride".

Answer: Thank you for pointing this out. We have made the necessary correction.

4) For "Silyboranate Et₃SiB"Pin were prepared according to the procedures previously 1 and 4" should be the original paper, not the paper by the authors themselves.

Answer: Thank you for pointing this out. We have revised this accordingly.

Reviewer #2 (Remarks to the Author):

Prof Shibata et al have developed a novel method for the cross coupling of aryl fluorides with secondary amines using super-stoichiometric amounts of reagents Et₃SiBPin, KOtBu and R₂NH. The paper is presented in a logical manner and the prose is of a high degree of scholarship. The scope of the reaction includes mostly para-substituted aryl fluorides with arylamines, although the authors have extended the reaction to a few aliphatic amines as well as ortho- and meta-substituted aryl fluorides, albeit in reduced yields. Nevertheless, the authors have developed an impressive scope of reactivity involving multiple functional groups, EWGs/EDGs, and have applied their reaction to the synthesis of potentially high-value pharmaceutical targets. All newly synthesized molecules are well characterized.

However, the proposed mechanism is likely incorrect, and a much simpler alternative is available that is also consistent with their mechanistic findings. The authors propose a SET event to produce intermediate

B (Fig 3), which is a dianionic boryl, as well as $\text{Et}_3\text{Si}\cdot$ and ${}^t\text{BuO}\cdot$. There are a number of things troubling with this – surely the two radical components would immediately recombine to produce stable $\text{Et}_3\text{SiO}{}^t\text{Bu}$, even at least to some degree if they were present in the reaction? The authors however state that only Et_3SiH is detected in the reaction mixture. Furthermore, intermediate B looks atrociously unstable, and to my knowledge a dianionic boryl $[(\text{RO})_3\text{B}]^{2-}$ has no precedence in low-valent boron chemistry, and similar species can only be stabilized by the presence of strong pi-acceptors, such as cyanide. *There seems to be a typo at intermediate C where an extra hydrogen atom has appeared.* While the intermediacy of an amido radical is indicated from the control experiments, the 4-component radical cascade involving double radical transfer from intermediate B seems very dubious.

There is a much simpler alternative:

This seems to be fully consistent with the observed results and the structures proposed have a good deal of precedence in Frustrated radical pair chemistry. This would also be easy to test – combine KNR_2 with $\text{BPin}(\text{OtBu})$ and see if this performs nucleophilic substitution on aryl fluorides. Also, the mechanism in Fig 3 is drawn as a circle, which usually indicates catalysis. This is not catalysis - this reaction is superstoichiometric in every reagent. To draw it as a circle seems misleading. Once these corrections are made I would find the article suitable for publication. Other than the mechanistic work, the conclusions of the

paper are well substantiated, the method described is simple, novel, seems to be fairly generally applicable, and is a useful addition to the chemical toolbox to make C-N bonds.

Response: Thank you very much for your excellent and very ideal reaction mechanism !!! We are very pleased about your suggestion.

As per your suggestion, we further attempted a few experiments to confirm the possibility of an alternative mechanism. By treating *N*-methylaniline with a KHMDS solution (0.5 M in toluene) under room temperature, we obtained a potassium salt of the amide (KNPhMe) as a solid precipitate. Thereafter, we examined two experiments. First, we attempted the reaction of KNPhMe with **1a** in the presence of ^tBuOBpin in triglyme under room temperature for 24 hours. However, no reaction occurred. Next, we attempted the reaction of KNPhMe and the Cl-substituted aryl fluoride **1m** under the same conditions, which only produced a small amount of S_NAr by-product at the Cl position (approximately 5%). In both cases, the starting materials **1a** and **1m** largely remained unreacted. Therefore, the reaction should not include the generation of KNR₂ (NR₂ anion). Moreover, if KNR₂ (i.e., an anion of NR₂) is generated, substrate **1i** and **1o** (Cl-), **1p** (Br-), and **1q** (CN-) should be easily undergone the S_NAr process, but not. The desired products (**4ia**, **4oa**, **4pa**, **4qa**, **4oi**, and **4qi**) were obtained in high selectivity and yield (Table 2). In addition, the S_N2 type pathway is ruled out based on the comparisons of alkyl-F vs alkyl-Br, and alkyl-Cl (mentioned above, Figures 3i, j and k in the revised manuscript, for the response to the reviewer 1). Based on these new facts, and thanks to the referee's excellent suggestion, we propose the modified reaction mechanism based on your suggested simpler mechanism, including the radical of NR₂, not the pure anion of NR₂ (Figure 3i in the revised manuscript). Thank you very much once again.

Reviewer #3 (Remarks to the Author):

This paper by Shibata and co-workers deals with the silylboronate-mediated selective defluorinative cross-coupling of organic fluorides with secondary amines via a transition-metal-free strategy and is strongly based on some previous reports. In particular, the authors recently published a paper (*Org. Lett.* **2022**, *24*, 508) whose title (Etherification of Fluoroarenes with Alkoxyboronic Acid Pinacol Esters via C–F Bond Cleavage) clearly indicates the close relation with the present paper. In fact, the only essential difference is the use of amines for the generation of *N*-radical species, instead of an alkoxy radical derived from RO-Bpin in the mentioned communication. What is indeed very surprising is the fact that the authors omitted the citation of this paper. Therefore, the novelty in this paper lies just in the development of an extension of this known process, which is indeed a valuable contribution, but is far from meeting the strict novelty criteria for publication in a leading journal such as Nature Communications.

Response: Before we submitted this manuscript to *Nature Communications*, the etherification paper (*Org. Lett.* **2022**, *24*, 508) had not been published. Therefore, we planned to cite this paper in our current manuscript during the revision process. Moreover, the concepts of the reactions reported in both papers are entirely different from each other. The present work (this manuscript) describes an Si radical-mediated cascade reaction that generates an NR_2 radical from NHR_2 at room temperature. In contrast, the reaction reported in our suggested work (*Org. Lett.* **2022**, *24*, 508) makes the use of RO-BPin (not ROH) at high temperature (110 °C) under strong basic conditions (KHMDS) and does not use any Si species, and cannot use ROH directly. We have now cited this paper in the revised manuscript as per your suggestion.

REVIEWER COMMENTS

Reviewer #1 (Remarks to the Author):

After reviewing the revised manuscript, since the problems previously pointed out seem to have been appropriately corrected. The reviewer thus concluded that this manuscript could be accepted for publication in Nature Communications if the following points are corrected.

1) The structural formula of products 4ka-4ta in Table 2 is incorrect. When the structural formula is written as compounds 4af-4ak, compound 4ka has a naphthyl group, so the structural formula should be drawn separately. For other products, for example, "R = 4-Me (Tol)" should be used for 4la. Also, "Ph" means C₆H₅, so it should correctly be written as "C₆H₄" or "C₆H₃" when there are two substituents.

2) Page 10, line 212: "(Table 1, entries 14-16)" is incorrect for "(Table 1, entries 16-18)".

Prof Shibata et al have developed a borane-catalysed method for the cross coupling of aryl fluorides with secondary amines using super-stoichiometric amounts of reagents Et_3SiBPin , KOtBu and R_2NH . The paper is presented in a logical manner and the prose is of a high degree of scholarship. The scope of the reaction includes mostly para-substituted aryl fluorides with arylamines, although the authors have extended the reaction to a few aliphatic amines as well as ortho- and meta-substituted aryl fluorides, albeit in reduced yields. Nevertheless, the authors have developed an impressive scope of reactivity involving multiple functional groups, EWGs/EDGs. All newly synthesized molecules are well characterized, and the conclusions are generally well-substantiated.

However, while the authors have used their method for the preparation of a few high-value products, its unlikely that the method would be applicable to many late stage functionalizations or the preparation of deuterated derivatives due to the need for super-stoichiometric amine (3 equiv with respect to Ar-F). Using chiral or deuterated amines in these reactions would be very inefficient.

RSC Adv., 2015, 5, 7035 details a similar uncatalyzed reaction involving the amination of C-F bonds, though the scope of fluoroarenes in the present study is much larger. This paper should at least be cited. Additionally:

<https://chemistry-europe.onlinelibrary.wiley.com/doi/10.1002/chem.201604098>

Journal of Fluorine Chemistry, 2017, 204, 59

Eur. J. Org. Chem., 2017, 4300

all detail the TM-free formation of C-N bonds from activated and un-activated C-F bonds and are not cited. A clearer picture of the contributions of other researchers and the advantage of the present method needs to be made.

Line 20 – it seems odd to address the desirable properties of C-F bonds when they are being used only as reactants in this study. They certainly do present a unique challenge, and more discussion of previous efforts to functionalize them (like the studies linked above), and how this study fits into the picture would be more appropriate.

Line 39 – I am unsure of the point being made here – is it not advantageous to develop methods for the functionalization of electron rich aryl fluorides?

Line 52 – “Additionally” not “besides”

Line 60 – save for 5ea (20% yield), the present study is also restricted to benzylic $\text{Csp}^3\text{-F}$ fluorides. The subsequent sentence about the present method being suitable for inert sp^3 C-F amination is certainly overstated.

Line 73- carried “out” in

Line 187. The radical derived from H-atom abstraction of methylaniline is not cationic, and this statement seems contrary to the proposed mechanism.

Line 189 – without characterizing the products of this reaction it is impossible to support or disprove radical ring-opening. This piece of data is inconclusive.

Line 208 – leading references on Frustrated Radical pair chemistry should be included

Reviewer #2 (Remarks to the Author):

Once these corrections/additions have been made I would find the article suitable for publication in Nature Communications

RESPONSE TO REVIEWERS:

We thank you for your thoughtful suggestions and insights, which have enriched the manuscript and produced a better and more balanced account of the research.

Reviewer #1 (Remarks to the Author):

After reviewing the revised manuscript, since the problems previously pointed out seem to have been appropriately corrected. The reviewer thus concluded that this manuscript could be accepted for publication in Nature Communications if the following points are corrected.

1) The structural formula of products 4ka-4ta in Table 2 is incorrect. When the structural formula is written as compounds 4af-4ak, compound 4ka has a naphthyl group, so the structural formula should be drawn separately. For other products, for example, "R = 4-Me (Tol)" should be used for 4la. Also, "Ph" means C₆H₅, so it should correctly be written as "C₆H₄" or "C₆H₃" when there are two substituents.

Response: Thank you for pointing this out. We are sorry to make you confused on the structural formula of products **4ka-4ta**. We have revised accordingly. For the substituent group in structure of **4ka**, it can be abbreviated as "Ar = 1-Naphthyl" (*Nat. Commun.* **2022**, **13**, **7649**).

2) Page 10, line 212: "(Table 1, entries 14-16)" is incorrect for "(Table 1, entries 16-18)".

Response: Thank you for pointing this out. We have revised this mistake.

Reviewer #2 (Remarks to the Author):

Prof Shibata et al have developed a borane-catalysed method for the cross coupling of aryl fluorides with secondary amines using super-stoichiometric amounts of reagents Et₃SiBPin, KOtBu and R₂NH. The paper is presented in a logical manner and the prose is of a high degree of scholarship. The scope of the reaction includes mostly para-substituted aryl fluorides with arylamines, although the authors have extended the reaction to a few aliphatic amines as well as ortho- and meta-substituted aryl fluorides, albeit in reduced yields. Nevertheless, the authors have developed an impressive scope of reactivity involving multiple functional groups, EWGs/EDGs. All newly synthesized molecules are well characterized, and the conclusions are generally well-substantiated.

Response: Thank you for your high evaluation again.

However, while the authors have used their method for the preparation of a few high-value products, it is unlikely that the method would be applicable to many late stage functionalizations or the preparation of deuterated derivatives due to the need for super-stoichiometric amine (3 equiv with respect to Ar-F). Using chiral or deuterated amines in these reactions would be very inefficient.

Response: Thank you for raising this critical issue. Yes, I understand and it is the next challenge. Despite these drawbacks, the mild reaction conditions (room temperature) offer significant advantages over previous methods (over 100°C in harsh basic conditions; see following citations for example), resulting in realized late stage functionalizations. Although an excess of deuterated methylaniline derivatives or chiral amines is required, the unreacted amines can be recovered during column chromatography purification. This allows us to maximize the use of expensive

amine precursors and save costs. Your valuable comments lead us to the next challenge. Thank you very much.

RSC Adv., 2015, 5, 7035 details a similar uncatalyzed reaction involving the amination of C-F bonds, though the scope of fluoroarenes in the present study is much larger. This paper should at least be cited.

Additionally:

<https://chemistry-europe.onlinelibrary.wiley.com/doi/10.1002/chem.201604098>

Journal of Fluorine Chemistry, 2017, 204, 59

Eur. J. Org. Chem., 2017, 4300

all detail the TM-free formation of C-N bonds from activated and un-activated C-F bonds and are not cited. A clearer picture of the contributions of other researchers and the advantage of the present method needs to be made.

Response: Thank you for your suggestions. We have cited these references with appropriate sentences in the introduction.

Line 20 – it seems odd to address the desirable properties of C-F bonds when they are being used only as reactants in this study. They certainly do present a unique challenge, and more discussion of previous efforts to functionalize them (like the studies linked above), and how this study fits into the picture would be more appropriate.

Response: Thanks for pointing that out. Based on your valuable suggestion, I added more explanation of C-F bond functionalization (like the studies you suggested in the above comments) in the **middle** position of the introduction part (**not the beginning** on line 20).

PS: I want to keep line 20 as it is. Please understand because I want to start the story with the importance of organofluorine compounds. Our motivation for doing this work begins with the abundance of organofluorine compounds. In the last century, fluorinated compounds were precious and expensive, even simple compounds. Thus, the removal of fluorine from organofluorine compounds was relatively meaningless. Therefore, many people have made efforts to develop an efficient synthetic method for organofluorine compounds. However, nowadays, due to the extraordinary efforts of synthetic chemists, organofluorine compounds have become abundant not only for simple structures but also for complex molecules such as pharmaceuticals and agrochemicals. Thus, another chemistry of activating the C-F bond is going to be valid. Therefore, we would like to start the story in this manuscript with the why of organofluorine compounds.

Line 39 – I am unsure of the point being made here – is it not advantageous to develop methods for the functionalization of electron rich aryl fluorides?

Response: Thank you for your comments. My sentences should have been clearer. I modified the sentences accordingly.

My meaning is as follows;

Traditional methods require strong electron-withdrawing substituents such as NO₂ or multiple F moieties. The new methods realized the reaction of electron-rich aryl fluorides with electron-donating (OMe) moiety. However, the new methods are unsuitable for electron-poor, electron-neutral and non-substituted aryl fluorides. The OMe group must be required. On the other hand, our method accepts all substitutions, including electron-rich, electron-poor and neutral and non-substituted (H) aryl fluorides. I modified the sentences accordingly.

Line 52 – “Additionally” not “besides”

Response: We have revised.

Line 60 – save for **5ea** (20% yield), the present study is also restricted to benzylic Csp³-F fluorides. The subsequent sentence about the present method being suitable for inert sp³ C-F amination is certainly overstated.

Response: Thank you for pointing this out. I agree with our overstating, and we rephrased this sentence accordingly.

Line 73- carried “out” in.

Response: Thank you for pointing this out. We have revised this mistake.

Line 187. The radical derived from H-atom abstraction of methylaniline is not cationic, and this statement seems contrary to the proposed mechanism.

Response: Thank you for pointing this out. The related sentences were mistaken. We have revised the related sentences accordingly.

Line 189 – without characterizing the products of this reaction it is impossible to support or disprove radical ring-opening. This piece of data is inconclusive.

Response: Thanks for pointing this out. We have examined the additional experiments and obtained information about radical ring-opening phenomena.

1. First, the reaction of **1a** with **3w** under standard conditions did not give the desired product **4aw** (already mention in the last version). Interestingly, most of **1a** were recovered (about 90%). However, **3w** disappeared completely, while ¹H-NMR showed complex mixtures. Thus, these are difficult to assign.
2. We then treated only **3w** under standard conditions, resulting in the complex mixture and **3w** disappearing. This fact is consistent with the literature that the generated aminyl radical **3w'** spontaneously transforms into the ring-opening primary carbon radical **3w''** (J. Am. Chem. Soc. 102, 328–331 (1980); Chem. Soc. Rev. 51, 7344–7357 (2022); Acc. Chem. Res. 49, 1957–1968 (2016)). Thus, the amine radical **3w'** does not exist anymore, and the resulting ring-opening primary radical **3w''** is unstable and decomposes into the complex mixture.
3. On the other hand, treatment of **3w** under standard conditions without Et₃SiBpin resulted in the recovery of **3w**. This is reasonable since the anion **3w'''** is stable and **3w** was recovered by the work-up process.

These results (from 1-3) support the radical process over the anionic process. We added these results in the revised manuscript

Line 208 – leading references on Frustrated Radical pair chemistry should be included

Response: Thank you for pointing this out. We have added references accordingly.

Once these corrections/additions have been made I would find the article suitable for publication in Nature Communications

Response: Thank you very much for your valuable suggestions to improve our chemistry.